# The Quality of Anticoagulation Therapy among Warfarin-Treated Patients with Atrial Fibrillation in a Primary Health Care Setting

**DOI:** 10.3390/medicina55010015

**Published:** 2019-01-15

**Authors:** Gediminas Urbonas, Leonas Valius, Gintarė Šakalytė, Kęstutis Petniūnas, Inesa Petniūnienė

**Affiliations:** 1Department of Family medicine, Medical Academy, Lithuanian University of Health Sciences, LT-50161 Kaunas, Lithuania; leonas.valius@fc.lsmuni.lt (L.V.); petniunaskestutis@gmail.com (K.P.); i.anulyte@gmail.com (I.P.); 2Department of Cardiology, Medical Academy, Lithuanian University of Health Sciences, LT-50161 Kaunas, Lithuania; gintare.sakalyte@fc.lsmuni.lt

**Keywords:** atrial fibrillation, anticoagulation, INR, TTR, warfarin, primary care

## Abstract

*Background and objectives:* Long-term therapy with oral anticoagulants is recommended for stroke prevention in patients with atrial fibrillation (AF). This study evaluated the quality of anticoagulation therapy among warfarin-treated AF patients in selected primary health care centres in Lithuania. *Materials and Methods*: This was a retrospective study conducted in nine primary health care centres in Lithuania. Existing medical records of randomly selected adult patients with AF who were treated with warfarin for at least 12 months were reviewed and analysed. Physicians’ decisions to adjust warfarin dose were considered as consistent with the approved warfarin posology if warfarin dose was increased in case of international normalized ratio (INR) <2.0, decreased in case of INR >3.0 or unchanged in case of INR within 2.0 to 3.0. *Results:* The study population included 406 patients. The mean duration of treatment with warfarin was 5.4 years. The median number of INR measurements per patient per year was 8.0. More than half (57.3%) of available INR values were outside the target range, with 13.6% INR values being above 3.0 and 43.7% INR values—below 2.0. The median time in therapeutic range (TTR) was 40.0%; only 20% of patients had TTR of ≥65%. In about 40% of the cases with INR values outside the target range, no dose corrections were implemented. About 27% of decisions on warfarin dose adjustment were not consistent with the recommended warfarin posology. The median number of INR measurements was lower among patients living in urban areas, while the median TTR was significantly higher in urban patients than in rural patients. In the multivariate regression model, gender, HAS-BLED score and warfarin treatment duration were associated with a TTR of ≥65%. *Conclusions*: Anticoagulation control is suboptimal in routine clinical practice with a median TTR of 40%. Our findings suggest that there might be a room for improvement of anticoagulation control in primary care.

## 1. Introduction

Atrial fibrillation (AF) is the most common disorder of heart rhythm. There were approximately 8.8 million adults with AF in the European Union in 2010 [1]. The prevalence of AF is increasing, mainly because of population aging, and it is estimated that the number of patients with AF will be doubled by 2060 [1]. Atrial fibrillation is associated with significant morbidities and mortality due to stroke and peripheral arterial embolism [2]. The risk of stroke is increased four- to five-fold in patients with AF [3].

Clinical guidelines for stroke prevention in patients with AF recommend long-term therapy with oral anticoagulants [4]. Despite the availability of a new class of anticoagulants, known as non-vitamin K oral anticoagulants (NOACs), the vitamin K antagonist (VKA), warfarin, remains the preferred anticoagulant for many patients [5]. The main reasons include lower warfarin price, contraindications and lower adherence to NOACs, as well as concerns for increased risk of myocardial infarction with NOACs [5]. Results from randomized controlled trials support the use of warfarin in patients with AF who are at moderate or high risk of stroke. Assuming a baseline risk of 45 strokes per 1000 patient-years, warfarin could prevent 30 strokes at the expense of only six additional major bleeds [6]. In patients with AF, adjusted-dose warfarin reduced stroke by approximately 60% compared with no treatment and by approximately 40% compared with antiplatelet therapy [7]. However, the use of warfarin is challenging because of its multiple interactions and narrow therapeutic index. Warfarin therapy requires continuous dose adjustment to maintain the international normalized ratio (INR) between 2.0 and 3.0. INR greater than 3.0 poses a higher risk of bleeding and INR less than 2.0 may increase a risk of thromboembolism [2,8]. It is recommended to determine INR at least weekly during initiation of warfarin therapy and at least monthly when anticoagulation is stable (INR in range) [4].

The time a patient spends within the therapeutic range (time in the therapeutic range; TTR) is accepted as a measure of anticoagulation quality. A 10% increase in time spent out of TTR is associated with a 29% increase in the risk of mortality, and 10 to 12% increase in the risk of an ischemic stroke and other thromboembolic events [9]. In the Global Anticoagulant Registry in the FIELD–Atrial Fibrillation (GARFIELD-AF) study, patients with TTR <65% had a 2.6-fold higher risk of stroke, 1.5-fold higher risk of major bleeding, and 2.4-fold higher risk of all-cause mortality [10]. A recent meta-analysis of 31 studies confirmed that increasing mean TTR is associated with a lower rate of both major bleeding and stroke/systemic embolism [11].

In this study, we aimed to evaluate the quality of anticoagulation therapy among warfarin-treated AF patients in selected primary health care centres (PHCCs) in Lithuania.

## 2. Materials and Methods

This was a retrospective study conducted in nine PHCCs (serving a total of 195,073 people) in Lithuania. The study was approved by the Ethics Committee of Lithuanian University of Health Sciences (issue number, BEC-LSMU(R)-21, 5 December 2016) and by the Kaunas Regional Biomedical Research Ethics Committee (issue number, BE-2-74, 8 October 2018). Six PHCCs were located in towns (Kaunas Dainava outpatient Clinic, Siauliai Central Outpatient Clinic, Kedainiai Primary Health Care Centre, Clinic of Family Medicine of Hospital of Lithuanian University of Health Sciences Kauno Klinikos, Saules Family Medicine Centre, Vita Longa Family Medicine Centre) and three PHCCs in rural areas (Kaltinenai Primary Health Care Centre, Rekyva Outpatient Clinic, Babtai Family Medicine Centre). Between January 2016 and January 2017, existing medical records of randomly selected adult patients with non-valvular AF who were treated with warfarin for at least 12 months were reviewed. In each PHCC, all patients with AF diagnosis (code I48 according to the International Classification of Diseases, 10th revision) were identified. Patients fulfilling inclusion criteria were selected in alphabetic order until the predefined number of patients was achieved in each PHCC. The data collected included details on health care centre (private or public), patient sociodemographic (gender, age, social status, residential area, distance to health care centre), concomitant medications with a potential of interaction with warfarin, risk factors for stroke and bleeding, INR results, warfarin treatment duration and warfarin dose (including dose corrections).

The HAS-BLED (Hypertension, Abnormal renal/liver function, Stroke, Bleeding history or predisposition, Labile INR, Elderly, Drugs/alcohol concomitantly) score was calculated by assigning 1 point each for the presence of arterial hypertension, renal or hepatic impairment, history of stroke, history of bleeding, labile INR, age ≥ 65 years, alcohol or drug abuse; 2 points were assigned if both renal and hepatic impairment or alcohol and drug abuse were presented. Patients were classified into two groups of bleeding risk according to the total HAS-BLED score: low risk (score ≤2) or high risk (score ≥3) [12].

The CHA2DS2-VASc (Congestive heart failure, Hypertension, Age ≥ 75 years, Diabetes, Stroke, Vascular disease, Age 65–74 years, Sex category) score was calculated by assigning 1 point each for the presence of heart failure, arterial hypertension, diabetes mellitus, vascular disease, age 65–74 years or female sex and 2 points for a history of stroke or transient ischemic attack or age ≥ 75 years [12].

Individual TTRs were determined by calculating the fraction of all INR values that were within the therapeutic range (i.e., the number of INRs in the range between 2.0 and 3.0 divided by the total number of INR tests). We stratified patients into two groups based on TTR level: TTR ≥65% (good control) and TTR <65% (poor control) (a cut-off value of <65% is used in the national conditions for reimbursement of NOACs in Lithuania as an indicator of poor anticoagulation control) [13].

Physicians’ decisions to adjust warfarin dose were analysed. The decision was considered as consistent with the approved warfarin posology [14] if warfarin dose was increased in case of INR <2.0 or decreased in case of INR >3.0 or unchanged in case of INR within 2.0 to 3.0. The decision was considered as not consistent if warfarin dose was not changed in case of INR out of the target range (i.e., 2.0 to 3.0) or increased in case of INR >3.0 or decreased in case of INR <2.0.

Patients were grouped according their ability to work as follows: fully able to work (a person who has no restrictions to work because of age or a heath condition), retired (a person who left his or her job after reaching retirement age), disabled (a person unable to work because of a health condition) and not self-sufficient (a person unable to live without a permanent help of a caregiver).

Categorical data were presented as a count (n) and percentage (%). Continuous variables were presented as mean (standard deviation; SD) or median (interquartile range; IQR). We used the parametric Student *t*-test and nonparametric Mann-Whitney U for comparison of continuous variables between two independent samples. Spearman’s rank correlation coefficient was used to assess correlation between CHA2DS2-VASc score and HAS-BLED score, and between the intensity of INR measurements and TTR. We conducted univariate and multivariate logistic analyses to identify factors that were independently associated with values of TTR ≥65.0%. The dependent variable was TTR (TTR <65.0% versus TTR ≥65.0%). The independent variables used in the univariate analysis were age, type of health care centre (private owned), gender (male), ability to work (retired, disabled, or not self-sufficient versus fully able to work), place of residence (rural), distance to health care centre, HAS-BLED score, CHA2DS2-VASc score, number of INR measurements per year, duration of warfarin treatment and use of medicines interacting with warfarin. Multivariate logistic regression was employed using the enter method, i.e., all variables that were significant in univariate analysis were entered into the equation at the same time. To build a final model, other variables were entered to test their contribution to the regression equation. Statistical differences were interpreted at 5% (two-sided) significance level. Data were analysed using the statistical software package SPSS version 20.

## 3. Results

The study population included 406 patients. Mean (SD) age of the patients was 75.2 (9.4) years, 55.7% were female and 70.9% were retired. The duration of treatment with warfarin ranged from 1 to 20 years. The mean treatment duration did not differ between males and females (5.2 (3.3) and 5.5 (4.0) years, respectively; *p* = 0.934). A total of 309 patients (76.1%) were taking concomitant medications with a known potential of interaction with warfarin. Baseline sociodemographic and clinical characteristics are summarized in Table 1.

The majority of patients (89.6%) had a high risk of stroke (CHA2DS2VASc score ≥3), and 63.5% of patients were at high risk of bleeding (HAS-BLED score ≥3) (Table 1). The HAS-BLED score and CHA2DS2-VASc score showed a moderate correlation (Spearman’s rho: 0.476; *p* < 0.001).

The median (IQR) number of INR measurements per patient per year was 8.0 (6.0–10.0). More than half (57.3%) of available INR values were outside the target range, mostly below 2.0. The median TTR was 40.0%. Only 20% of patients had TTR ≥65%. Within one-year observational period, 31% of patients had at least two INR values below 1.5 (Table 2). The correlation between the number of INR measurements and TTR was insignificant (Spearman’s rho: 0.089; *p* = 0.074).

The most common corrective actions taken by physicians were warfarin dose increases in the case of INR <2.0, and dose reductions in the case of INR >3.0. However, in about 40% of cases of INR outside the target range, no dose corrections were implemented (Table 2). There were 2423 (73.16%) decisions on warfarin dose adjustment consistent with the recommended warfarin posology and 889 (26.84%) inconsistent decisions. The proportion of decisions consistent with the recommended posology was significantly greater in PHCCs located in rural areas to compare to urban areas (77.8% versus 72.0%; chi square *p* < 0.01).

The control of anticoagulation as expressed by the median number of INR measurements per patient was not significantly different between men and women, younger and older patients, among patients with different ability to work or bleeding risk (Table 3). Median TTR was significantly lower in patients with high bleeding risk (36.4%) as compared to low risk patients (55.6%) (Table 3). Among patients taking concomitant medicines that are known to increase the effect of warfarin, the median number of INR measurements was lower as compared to patients taking no medicines with a potential of interaction with warfarin. The median number of INR measurements was lower among patients living in urban areas, while no differences were observed in relation to distance to health care centre. In contrast, median TTR was significantly higher in urban patients than in rural patients (Table 3).

In the univariate analysis, higher HAS-BLED scores and CHA2DS2-VASc scores were significantly associated with lower odds ratios to attain TTR of ≥65.0% (Table 4). In the multivariate logistic analysis (Table 5), gender (*p* = 0.041), HAS-BLED score (*p* < 0.001) and duration of warfarin treatment (*p* = 0.029) were significantly associated with TTR of ≥65.0%. Male gender and higher HAS-BLED score were associated with decreased odds ratios of good anticoagulation. Each increase in HAS-BLED score by 1 point decreased the odd of having TTR of ≥65.0% by 0.42, whereas each additional year of warfarin treatment increased these odds ratio by 1.08.

## 4. Discussion

In our cohort of patients taking warfarin as an oral anticoagulant, only 20% of the patients had a TTR of ≥65%. Less than half (43%) of available INR values were within the therapeutic range of 2.0 to 3.0. Most of INR values out of the therapeutic range were due to inadequate anticoagulation (i.e., <2.0). We previously reported similar results in a smaller sample of patients [15]. Another Lithuanian study found that only 19% of the patients receiving anticoagulation therapy (mostly VKAs) were in the therapeutic range (INR value between 2.0 and 3.0) [16]. Although the population in that study was slightly different (i.e., patients with AF attending cardiologist consultation at the tertiary care centre), the most common reason of being outside therapeutic range was the same, i.e., INR <2.0. Other studies conducted in various countries also reported that patients outside the therapeutic INR range are mostly because of undercoagulation rather than overcoagulation [8,10,17]. It has been suggested that physicians might be reluctant to increase warfarin dose even in response to too low INR [17]. A specially designed retrospective study failed to identify the specific causes of over- and undercoagulation in patients on warfarin therapy. In that study, the common causes of undercoagulation included response to previous change in dosage (16.4%), noncompliance or dosing errors (16.3%) and initiation of therapy (15.6%). A combination of several factors (i.e., changes in drugs, medical condition, dietary vitamin K intake, alcohol use and activity level) accounted for only 15.1% of INRs below 2.0. The cause of undercoagulation remained unknown in 29.7% of INR values below 2.0 [18].

The quality of anticoagulation with VKAs, as measured by TTR, varies considerably between studies, study sites and countries [8,19,20,21,22,23]. Patients treated in specialist care setting have higher TTR compared to primary care [8,24,25]. Currently, there is no consensus whether a critical level of TTR exists below which anticoagulation therapy is ineffective. A post hoc analysis of a large randomized trial of oral anticoagulation therapy versus clopidogrel plus aspirin indicated a critical value for TTR of 65% [20]. In the GARFIELD-AF study, patients with TTR <65% had a 2.6-fold higher risk of stroke, 1.5-fold higher risk of major bleeding and 2.4-fold higher risk of all-cause mortality [10]. In a UK study, stroke risk in patients with TTR of ≥70% was reduced by 79% compared to patients with TTR <30% [26]. A retrospective, multicentre cohort study based on Swedish registries suggested that individual TTR should be maintained at ≥70% [27]. In a recent Finish nationwide study, outcomes in AF patients taking warfarin continued to improve with increasing TTR values up to TTR ≥ 80% [28]. In international guidelines, the recommended cut-off of TTR ranges from 60% to 70% [12,29,30,31]. The national conditions for reimbursement of NOACs in Lithuania allow a reassessment of warfarin-based anticoagulation therapies if TTR is <65% [12].

The only patient characteristics associated with TTR in our multivariate regression model were gender, HAS-BLED score and the duration of warfarin treatment. Several studies demonstrated that poor anticoagulation quality is associated with female gender [15,32,33] and anticoagulation quality is deteriorated over time [9,24,32,34]. In contrast to previously reported results [15,32,33], in our study, males had a higher risk of poor anticoagulation, suggesting that there could be some important confounding factors among male patients. It has been reported that patients’ characteristics only explain 2–4% of the variation in individual’s TTR in patients starting warfarin after the first diagnosis of AF or venous thromboembolism [35]. In patients with high quality TTR (≥70%) in an initial 6 months period on warfarin, patients’ characteristics did not strongly predict TTR in the second 6 months period, although heart failure moderately increased the risk of TTR deterioration [36]. In the US Outcomes Registry for Better Informed Treatment of Atrial Fibrillation (ORBIT-AF study, renal dysfunction, advanced heart failure, frailty, prior valve surgery and higher risk for bleeding (ATRIA score) or stroke (CHA2DS2-VASc score) was significantly associated with lower TTR [23]. The Veterans AffaiRs Study to Improve Anticoagulation (VARIA) found that hospitalizations, alcohol abuse, substance abuse, cancer and dementia were particularly strong predictors of poor control and that patient-level predictors of anticoagulation control were different at the initiation of warfarin therapy than at the later periods [32]. People with higher warfarin awareness (i.e., patients’ knowledge of warfarin’s effect and food-drug interactions) had higher TTR levels [37,38].

The unexpected finding in our study was less intense INR monitoring in patients taking concomitant medicines that are known to increase the effect of warfarin. We had no information that could explain such results. However, it might be that such patients are taking more concomitant medicines because of more severe conditions, and therefore, they are not able to attend a physician’s office for blood tests regularly.

In addition to patient-level characteristics, clinical skill in warfarin dosing decisions (i.e., extent to which warfarin dosing is consistent with a simple algorithm, which specified no dose change if the INR is in range and dose changes if the INR is out of range) has been shown to be an important determinant of TTR and the composite clinical outcome of stroke, systemic embolism and major bleeding [22]. We found that warfarin doses were not adjusted in about 40% cases of INR outside the target range. Nearly 30% of dose adjustment decisions were not consistent with the recommended warfarin posology. In addition, our patients had a median of 8.0 INR measurements per year, which is less than the number reimbursed by the National Health Insurance Fund (i.e., 12 INR measurements per year). INR monitoring may not be optimal for a variety of reasons, including too rare patients visits, requirements to receive a formal referral for a blood test, lack of patients’ awareness of the importance of coagulation control, etc.

The findings of this study might implicate that possibilities for improvement of anticoagulation management exist in primary health care settings. However, we only analysed dose adjustment-related actions, which followed INR measurement and had no information on other factors (e.g., concomitant diseases) that might have determined physicians’ decisions regarding warfarin dose adjustments. Of note, it is not recommended to adjust warfarin doses when single INR values are slightly out of range in patients with previously stable therapeutic INRs [39].

This study revealed that physicians working at rural PHCCs more frequently adjusted warfarin dose consistently with the recommended posology. Moreover, the INR monitoring was more intensive among patients living in rural areas, while no differences were observed in relation to distance to health care centre. Thus, rural PHCCs, which also served less people than urban PHCCs, seem to provide more optimal patient care. Despite anticoagulation management efforts, the median TTR in rural patients was significantly lower than in urban patients, implicating that patient-related factors might be important determinants of anticoagulation quality.

We found that patients at high bleeding risk (as assessed using HAS-BLED score) had lower TTR than patients at low risk. In addition, HAS-BLED score was significantly associated with having TTR of ≥65.0% in a multivariate logistic regression model. In the univariate analysis, CHA2DS2-VASc score was also associated with TTR of ≥65.0%. Other studies also reported that increased baseline bleeding and stroke risk is associated with poor INR control [40] or lower TTR [23]. It has been suggested that such paradox when higher risk patients appear to receive less optimal care may result from the negative impact of comorbidities on TTR either through poorer medication adherence or biological variation in clotting factors [34].

The results of this study should be interpreted with caution because of several limitations. First, this study has limitations of a retrospective observational design. Retrieved data depended on the accuracy and completeness of medical records available at PHCCs. Secondly, the selected PHCCs may not represent all Lithuanian PHCCs and whole population of warfarin-treated patients with AF; therefore, the generalizability of study findings is limited. In addition, we determined TTR using the fraction of INR’s in range methodology, which is used in routine practice, whereas other published studies mostly used the Rosenthal linear interpolation method. However, the direct comparison of different methodologies used for measuring TTR did not distinguish the method that best reflects the quality of anticoagulation management [41].

## 5. Conclusions

This study confirmed that anticoagulation control is suboptimal in routine clinical practice with a median TTR of 40%. Although some of our findings suggest that there might be a room for improvement of anticoagulation control in primary care, a further nationwide study focusing on disease-, patient- and physician-related factors is needed to identify potential targets for actions.

## Figures and Tables

**Table 1 medicina-55-00015-t001:** Sociodemographic and clinical characteristics of the study population.

Characteristics	
Type of health care centre	
private, n (%)	93 (22.9)
public, n (%)	313 (77.1)
Number of people served per health care centre, mean (SD ^1^)	21,674.8 (6792.2)
Age, mean (SD), years	75.2 (9.4)
Gender	
male, n (%)	180 (44.3)
female, n (%)	226 (55.7)
Place of residence	
rural, n (%)	75 (18.5)
urban, n (%)	331 (81.5)
Distance to healthcare centre, mean (SD), kilometres	3.50 (4.62)
Ability to work	
fully able to work, n (%)	34 (8.4)
retired, n (%)	288 (70.9)
disabled, n (%)	50 (12.3)
not self-sufficient, n (%)	34 (8.4)
Warfarin treatment duration, mean (SD), years	5.4 (3.7)
Concomitant medications with known potential to interact with warfarin	
strengthening the effect of warfarin, n (%)	183 (45.1)
lowering the effect of warfarin, n (%)	126 (31.0)
HAS-BLED ^2^	
Total score, mean (SD)	2.85 (1.00)
Low bleeding risk (score ≤ 2), n (%)	146 (36.0)
High bleeding risk (score ≥ 3), n (%)	258 (63.5)
CHA2DS2-VASc ^3^ total score	
Mean (SD)	4.6 (1.6)
1, n (%)	7 (1.7)
2, n (%)	34 (8.4)
3, n (%)	45 (11.1)
4, n (%)	104 (25.6)
5, n (%)	106 (26.1)
6, n (%)	60 (14.8)
7, n (%)	31 (7.6)
8, n (%)	15 (3.7)
9, n (%)	3 (0.7)

^1^ Standard deviation; ^2^ Hypertension, Abnormal renal/liver function, Stroke, Bleeding history or predisposition, Labile INR, Elderly, Drugs/alcohol concomitantly; ^3^ Congestive heart failure, Hypertension, Age ≥ 75 years, Diabetes, Stroke, Vascular disease, Age 65–74 years, Sex category.

**Table 2 medicina-55-00015-t002:** Characteristics of anticoagulation control.

Characteristics	
International normalized ratio (INR)	
Total number of INR ^1^ measurements, n	3337
INR value, median (IQR ^2^)	2.2 (1.7–2.6)
INR within 2.0–3.0, n (%)	1424 (42.7)
INR <2.0, n (%)	1458 (43.7)
INR >3.0, n (%)	455 (13.6)
INR >8.0, n (%)	9 (0.3)
Patients with at least two INR values of <1.5	125 (30.8)
Time in therapeutic range (TTR)	
TTR ^3^, median (IQR), %	40.0 (25.0–60.0)
TTR <50%, n (%)	238 (58.6)
TTR 50–65%, n (%)	85 (20.9)
TTR ≥65%, n (%)	83 (20.4)
Warfarin dose adjustments	
INR <2.0	
Dose increased, n (%)	804 (55.8)
Dose decreased, n (%)	34 (2.4)
No dose correction, n (%)	602 (41.8)
INR >3.0	
Dose increased, n (%)	7 (1.6)
Dose decreased, n (%)	276 (61.1)
No dose correction, n (%)	169 (37.4)
TTR <65%	
Dose increased, n (%)	784 (29.6)
Dose decreased, n (%)	326 (12.3)
No dose correction, n (%)	1538 (58.1)
TTR ≥65%	
Dose increased, n (%)	56 (8.5)
Dose decreased, n (%)	37 (5.6)
No dose correction, n (%)	570 (86.0)

^1^ International normalized ratio; ^2^ Interquartile range; ^3^ Time in therapeutic range.

**Table 3 medicina-55-00015-t003:** Comparison of anticoagulation control in patient subgroups.

Variable	INR ^1^ Measurements Per Patient		TTR ^2^	
	Median (IQR ^3^)	*p* Value	Median (IQR)	*p* Value
Gender				
Men	8.0 (6.0–10.0)		40.0 (20.0–58.3)	
Women	8.0 (6.0–10.0)	0.807	40.0 (25.0–60.6)	0.360
Age				
<65 years	9.0 (6.3–10.0)		47.2 (28.9–66.7)	
≥65 years	8.0 (6.0–10.0)	0.316	40.0 (22.2–60.0)	0.172
HAS-BLED ^4^				
Low bleeding risk	8.0 (6.0–10.0)		55.6 (33.3–71.4)	
High bleeding risk	8.0 (6.0–10.0)	0.520	36.4 (20.0–50.0)	<0.001
Concomitant medications with potential of interaction				
none	9.0 (6.0–11.0)		40.00(16.7–58.3)	
strengthening the effect of warfarin	7.0 (6.0–10.0)	0.048	40.0 (22.2–60.0)	0.559
lowering the effect of warfarin	8.0 (6.0–10.0)	0.066	33.3 (20.2–58.6)	0.658
both strengthing and/or lowering the effect of warfarin	8.0 (6.0–10.5)	0.499	41.7 (25.0–60.0)	0.317
Ability to work				
fully able to work	9.0 (6.8–11.0)		47.7 (26.7–62.5)	
retired	8.0 (6.0–10.0)	0.192	40.0 (22.2–60.0)	0.197
disabled	8.0 (7.0–10.3)	0.663	43.6 (25.0–60.0)	0.474
not self-sufficient person	9.0 (6.0–11.0)	0.757	47.2 (23.8–64.4)	0.920
Distance to health care centre				
<5 km	8.0 (6.0–10.0)		40.0 (25.0–60.0)	
≥5 km	9.0 (6.0–10.0)	0.755	41.7 (19.1–61.3)	0.751
Place of residence				
rural	10.0 (7.0–11.0)		33.3 (20.0–54.6)	
urban	8.0 (6.0–10.0)	<0.0001	41.7 (25.0–60.0)	0.040

^1^ International normalized ratio; ^2^ time in therapeutic range; ^3^ interquartile range; ^4^ Hypertension, Abnormal renal/liver function, Stroke, Bleeding history or predisposition, Labile INR, Elderly, Drugs/alcohol concomitantly.

**Table 4 medicina-55-00015-t004:** Univariate logistic regression analysis for values of TTR ≥65.0%.

Variable	Odds Ratio	95% Confidence Interval	*p* Value
Age	0.98	0.96–1.01	0.163
Type of health care centre (private owned)	1.09	0.62–1.92	0.772
Gender (male)	0.74	0.45–1.22	0.236
Ability to work			
retired (versus fully able to work)	0.97	0.40–2.34	0.951
disabled (versus fully able to work)	0.96	0.33–2.85	0.947
not self-sufficient (versus fully able to work)	1.19	0.38–3.74	0.77
Place of residence (rural)	0.62	0.31–1.23	0.173
Distance to health care centre	0.97	0.91–1.04	0.419
HAS-BLED ^1^ score	0.46	0.35–0.61	<0.001
CHA2DS2-VASc ^2^ score	0.85	0.72–0.99	0.035
Number of INR measurements per year	0.96	0.87–1.05	0.346
Duration of warfarin treatment	1.06	0.99–1.12	0.090
Use of medicines interacting with warfarin (versus not using interacting medicine)	1.12	0.60–2.06	0.727

^1^ Hypertension, Abnormal renal/liver function, Stroke, Bleeding history or predisposition, Labile INR, Elderly, Drugs/alcohol concomitantly; ^2^ Congestive heart failure, Hypertension, Age ≥ 75 years, Diabetes, Stroke, Vascular disease, Age 65–74 years, Sex category

**Table 5 medicina-55-00015-t005:** Multivariate logistic regression analysis for values of TTR ≥65.0%.

Variable	Odds Ratio	95% Confidence Interval	*p* Value
Gender (male)	0.57	0.33–0.98	0.041
HAS-BLED ^1^ score	0.42	0.31–0.57	<0.001
Duration of warfarin treatment	1.08	1.01–1.15	0.029

^1^ Hypertension, Abnormal renal/liver function, Stroke, Bleeding history or predisposition, Labile INR, Elderly, Drugs/alcohol concomitantly.

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
