# Peer review of "The Quality of Anticoagulation Therapy among Warfarin-Treated Patients with Atrial Fibrillation in a Primary Health Care Setting"

_1010-660X, 2019, doi:10.3390/medicina55010015_

Round 1

Reviewer 1 Report

The study of Urbonas et al carries out a retrospective analysis of warfarin treatment in AF patients in a number of primary care institutes in Lithuania. They show that the majority of the patients are not in the therapeutic range and that patients are only in the therapeutic range around 40% of the time. Although these data are observational they illustrate the fact that closer care is needed when assessing the effectiveness of warfarin treatment.

In general the manuscript is well written, although there are a few minor grammatical errors that need looking at (eg line 58; 'a GARFIELD AF study, should be 'In the GARFIELD AF study')

The Authors also state that the patients were randomly selected from the whole population- perhaps i missed it but what % of the complete population does this represent- i believe this is important to discuss?

It is also mentioned that some practitioners do not alter their dosing even when the INR is not optimal. Were these practitioners from similar centres or were they equally distributed across all centres?

It is stated that warfarin remains treatment of choice in many patients despite the availability of other agents. I believe there should be some discussion on the reasons for this.

How is the data considered when the patient goes on concomitant medication during follow up?

Finally, although the Authors suggest further studies should be carried out, perhaps they should state how they would propose to carry out such studies (at least in brief)

Author Response

Point 1: In general the manuscript is well written, although there are a few minor grammatical errors that need looking at (eg line 58; 'a GARFIELD AF study, should be 'In the GARFIELD AF study')

Response 1: the manuscript has been reviewed once more and identified grammar errors have been corrected.

Point 2: The Authors also state that the patients were randomly selected from the whole population- perhaps i missed it but what % of the complete population does this represent- i believe this is important to discuss?

Response 2: Nine primary health care centers where study data were collected serve a total of 195,373 people (including children). Unfortunately, we have no information about the age and gender distribution in the whole population. Since the prevalence of AF varies with age and gender, we doubt the value of further discussion about the representativeness of study sample. The number of people served in participating health care centers has been added into “Experimental Section”.

Point 3: It is also mentioned that some practitioners do not alter their dosing even when the INR is not optimal. Were these practitioners from similar centres or were they equally distributed across all centres?

Response 3: The proportion of decisions consistent with the recommended posology was higher in PHCCs located in rural areas to compare to urban areas. This information has been added into “Results“ section; the corresponding discussion has been presented in „Discussions“ section (lines 341-347).  

Point 4: It is stated that warfarin remains treatment of choice in many patients despite the availability of other agents. I believe there should be some discussion on the reasons for this.

Response 4: the following sentence has been added into “Introduction” section: “The main reasons include lower warfarin price, contraindications and lower adherence to NOACs, as well as concerns for increased risk of myocardial infarction with NOACs”.

Point 5: How is the data considered when the patient goes on concomitant medication during follow up?

Response 5: the follow-up was not foreseen in this study, only data at the given moment were collected.

Point 6: Finally, although the Authors suggest further studies should be carried out, perhaps they should state how they would propose to carry out such studies (at least in brief).

Response 6: since the current study was conducted in the selected group of patients treated in few health care centers, further studies should attempt to cover the whole country. In addition, different types of AF should be considered. In “Conclusions” section, the last sentence has been modified as follows: “…a further nationwide study focusing on disease-, patient- and physician-related factors is needed to identify potential targets for actions”.

Reviewer 2 Report

In the given article, Urbonas and his colleagues, discussed the INR adjustment in primary care settings in retrospective analysis for a good sample size population of interest. The findings are interesting away from cardiology world and has clinical application. The inclusion of different AF parameters in the analysis with the risk scoring for both thromboembolism and bleeding is vital and incorporation with the TTR  was adequate as well. Despite the link to  what other conditions linked decisions changes, the regression analysis was ok to generate the hypothesis in the clinical settings. This paper will add the value of how PCP follow up their population and  help better to manage such population with better understanding of the disease dynamics as well. 

Author Response

Point 1: In the given article, Urbonas and his colleagues, discussed the INR adjustment in primary care settings in retrospective analysis for a good sample size population of interest. The findings are interesting away from cardiology world and has clinical application. The inclusion of different AF parameters in the analysis with the risk scoring for both thromboembolism and bleeding is vital and incorporation with the TTR  was adequate as well. Despite the link to  what other conditions linked decisions changes, the regression analysis was ok to generate the hypothesis in the clinical settings. This paper will add the value of how PCP follow up their population and  help better to manage such population with better understanding of the disease dynamics as well. 

Response 1:  the manuscript has been reviewed once more and identified grammar errors have been corrected.

Reviewer 3 Report

The paper titled “The Quality of Anticoagulation Therapy Among Warfarin-Treated Patients with Atrial Fibrillation in Primary Health Care Setting” performed a retrospective study conducted in nine selected primary health care centers (PHCCs) in Lithuania in order to evaluate the quality of anticoagulation therapy among warfarin-treated AF patients. In a study population that included 406 patients with a mean duration of treatment with warfarin of 5.4 years, the authors found that the median number of INR measurements per patient per year was 8.0. They also found that more than half of available INR values were outside the target range. The median time in therapeutic range (TTR) was 40.0, only 20% of patients had a TTR of ≥65%. In about 40% cases of INR outside the target range, no dose corrections were implemented. About 27% of decisions on warfarin dose adjustment were not consistent with the recommended warfarin posology.

This study shed light on some interesting findings that may have clinical importance. However, I have a few concerns regarding the presented data.

Major Comments 

1. The authors stated that the study participants/medical records were randomly selected. How was the randomization/selection done? Please describe this with more details. Of how many patients/medical records in total? The authors mentioned the number of patients served per health care center, could the authors specify if this number include only the warfarin-treated AF patients or not and whether this number corresponds only to the reporting period?

2.  In the result section, the authors presented statistical data regarding the type of the health care center (private/public), distance to health care center and ability to work, that weren’t discussed in the discussion section. Could the authors explain the rational of presenting these data if it will not be discussed later? The same comment goes to the urban patients vs the rural patients; the authors mentioned a TTR significantly higher in urban patients compared to rural patients. Although this data has been mentioned in the abstract as well as the result section, it hasn’t been mentioned at all in the discussion section.

Minor Comments

1. Line 39: “AF associated with significant…..”, should be replaced by “AF is associated with significant…”

2. Line 181: "... rather that overcoagulation..." should be replaced by "...rather than overcoagulation..." 

Author Response

Point 1: The authors stated that the study participants/medical records were randomly selected. How was the randomization/selection done? Please describe this with more details. Of how many patients/medical records in total? The authors mentioned the number of patients served per health care center, could the authors specify if this number include only the warfarin-treated AF patients or not and whether this number corresponds only to the reporting period?

Response 1: The number of people served per all 9 health care centers (line 72) and the description of patients selection has been described in more details (lines 83-85) in “Experimental Section”. The number of people served includes all “clients” of these health care centers (including healthy persons and children) and represented the situation of the reporting period.

Point 2:  In the result section, the authors presented statistical data regarding the type of the health care center (private/public), distance to health care center and ability to work, that weren’t discussed in the discussion section. Could the authors explain the rational of presenting these data if it will not be discussed later? The same comment goes to the urban patients vs the rural patients; the authors mentioned a TTR significantly higher in urban patients compared to rural patients. Although this data has been mentioned in the abstract as well as the result section, it hasn’t been mentioned at all in the discussion section.

Response 2: While planning this study, we assumed that these factors might be related with anticoagulation quality. That is why these data were presented and included into logistic regression model. Since the proportion of decisions consistent with the recommended posology was higher in PHCCs located in rural areas to compare to urban areas, the discussion on distance to health care center and urban/rural patients has been presented in „Discussions“ section (lines 341-347).  

Point 3: Line 39: “AF associated with significant…..”, should be replaced by “AF is associated with significant…”

Response 3: this mistake has been corrected.

Point 4: Line 181: "... rather that overcoagulation..." should be replaced by "...rather than overcoagulation..." 

Response 4: this mistake has been corrected.

Reviewer 4 Report

I would first congratulate the authors for their work. Atrial fibrillation is the most common arrhythmia worldwide and is associated with high stroke risk. Vitamin K antagonists are still widely used for stroke prevention, however many patients on such treatment do not control well the level of anticoagulation. For this reason any new information regarding this topic is clinically important. 

I have some some comments and suggestions about the current manuscript that could improve additionally its quality:

1.       Line 38:  Are there newer data about the prevalence of atrial fibrillation (AF)? The presented data reflect the situation in 2010, the last European guidelines on AF were published in 2016 and it is now the end of 2018…

2.       Line 38: Age is definitely a strong non-modifiable risk factor for AF, but you should  by all means mention also some other major risk factors (modifiable): arterial hypertension, ischemic heart disease, heart failure, valvular heart disease, post-cardiac surgery, chronic lung diseases, etc.

3.       Line 39: AF should not be abbreviated in the beginning of the sentence – write the whole words. In the same sentence add “is” after “Atrial fibrillation”

4.       Line 40: Please, specify that AF is risky for ischemic stroke and peripheral arterial embolism (or arterial embolic events). Thromboembolism is too general: this term suggests also pulmonary embolism (and the very venous thrombotic disease) - AF has no direct role for it.

5.       Line 63: “…selected primary health care centers”. Please, would you point out in the experimental section the criteria for selection of these centers? Are they all the primary health centers in these districts or just the ones that agreed to participate..? Or other criteria used..?

6.        Please, would you specify all inclusion and exclusion criteria for participation of a patient in your study?

7.       Regarding the CHA2DS2-Vasc and HAS-BLED scores – they are well-described and widely used, so according to me it is not necessary to describe in your article in details the risk factors they are based on and the points. You can just mention that you have used CHADS-Vasc score for estimation of the stroke (or thromboembolic) risk and the HAS-BLED score for the bleeding risk. You might leave text with the cut-off points of each score indicating the corresponding risk.

8.       Would you present any data about the prevalence of the different types of AF in you study population: new diagnosed, paroxysmal, persistent, long-standing persistent and permanent? It will be interesting and clinically important to see if there is any association between the type of AF and the other variables evaluated by you, such as the stroke risk, bleeding risk, TTR, etc.

9.   Do you have any data what percentage of your patients had valvular or non-valvular AF (for the same considerations as above)?

10.   Line 125: The lowest cut-off value of CHA2DS2-Vasc Score you have defined as “high risk” is 4. According to the current European guidelines: 0 score is “low” risk”, a  score 1 is “low-moderate” risk, score 2 is “moderate-high” risk. High stroke risk is defined at CHA2DS2-Vasc score ≥3.  If you add the patients with CHA2DS2 Vasc = 3 pts from your study to those with ≥ 4 pts the percentage of high-risk patients might be even higher…

11.   Please, when you present the standard deviation, use the symbol ±, not ( ) next to the mean value

12.   In the beginning of “Results”:  I suppose the difference in gender prevalence is not significant, but yet, please, show the p-value.

13.   Was there any gender difference in duration of treatment with warfarin?

14.   There is some repetition of data in the text before table 1 and in the very table 1 (for example for “gender”). The same for table 2. Please, avoid repetition of data in the text and the tables. 

Add also some p –values in tables 1 and 2, where relevant.

Author Response

Point 1:     Line 38:  Are there newer data about the prevalence of atrial fibrillation (AF)? The presented data reflect the situation in 2010, the last European guidelines on AF were published in 2016 and it is now the end of 2018…

Response 1: unfortunately, we haven’t found newer data about AF prevalence in Europe. More recent publications (see below) present data on AF prevalence referring to the same publication as we did.

Gorenek B et al (2017) European Heart Rhythm Association (EHRA)/European Association of Cardiovascular Prevention and Rehabilitation (EACPR) position paper on how to prevent atrial fibrillation endorsed by the Heart Rhythm Society (HRS) and Asia Pacific Heart Rhythm Society (APHRS). Europace. 2017 Feb 1;19(2):190-225

Lane DA et al (2017). Temporal Trends in Incidence, Prevalence, and Mortality of Atrial Fibrillation in Primary Care. J Am Heart Assoc. 2017 Apr 28;6(5).

White Paper on inequalities and unmet needs in the detection of atrial fibrillation (AF) and use of therapies to prevent AF‑related stroke in Europe. November 2018.

Point 2:        Line 38: Age is definitely a strong non-modifiable risk factor for AF, but you should  by all means mention also some other major risk factors (modifiable): arterial hypertension, ischemic heart disease, heart failure, valvular heart disease, post-cardiac surgery, chronic lung diseases, etc.

Response 2: this mistake has been corrected.

Point 3: Line 39: AF should not be abbreviated in the beginning of the sentence – write the whole words. In the same sentence add “is” after “Atrial fibrillation”

Response 3: this mistake has been corrected.

Point 4:     Line 40: Please, specify that AF is risky for ischemic stroke and peripheral arterial embolism (or arterial embolic events). Thromboembolism is too general: this term suggests also pulmonary embolism (and the very venous thrombotic disease) - AF has no direct role for it.

Response 4: this mistake has been corrected.

Point 5:     Line 63: “…selected primary health care centers”. Please, would you point out in the experimental section the criteria for selection of these centers? Are they all the primary health centers in these districts or just the ones that agreed to participate..? Or other criteria used..?

Response 5: data were collected at health care centers where the authors of the study worked.

Point 6:      Please, would you specify all inclusion and exclusion criteria for participation of a patient in your study?

 Response 6: This study included adult patients with non-valvular atrial fibrillation (AF) who were treated with warfarin for at least 12 months were reviewed. There were no other inclusion/exclusion criteria.

Point 7:      Regarding the CHA2DS2-Vasc and HAS-BLED scores – they are well-described and widely used, so according to me it is not necessary to describe in your article in details the risk factors they are based on and the points. You can just mention that you have used CHADS-Vasc score for estimation of the stroke (or thromboembolic) risk and the HAS-BLED score for the bleeding risk. You might leave text with the cut-off points of each score indicating the corresponding risk.

Response 7: We assume that the audience of this journal is rather wide; therefore, the explanation how the scores are calculated may be useful to specialists who are not familiar with these scores (e.g., dentists, nurses or pharmacists) or medical students.

Point 8:      Would you present any data about the prevalence of the different types of AF in you study population: new diagnosed, paroxysmal, persistent, long-standing persistent and permanent? It will be interesting and clinically important to see if there is any association between the type of AF and the other variables evaluated by you, such as the stroke risk, bleeding risk, TTR, etc.

Response 8: Unfortunately, we did not collected data on different types of AF.

Point 9:   Do you have any data what percentage of your patients had valvular or non-valvular AF (for the same considerations as above)?

Response 9: This study included only patients with non-valvular AF.

Point 10:  Line 125: The lowest cut-off value of CHA2DS2-Vasc Score you have defined as “high risk” is 4. According to the current European guidelines: 0 score is “low” risk”, a  score 1 is “low-moderate” risk, score 2 is “moderate-high” risk. High stroke risk is defined at CHA2DS2-Vasc score ≥3.  If you add the patients with CHA2DS2 Vasc = 3 pts from your study to those with ≥ 4 pts the percentage of high-risk patients might be even higher…

Response 10: Proportion of patients with CHA2DS2-Vasc score ≥3 has been presented in “Results” section. In Table 2, the proportions of patients with each value of this score have been presented instead of three stroke risk groups.

Point 11:  Please, when you present the standard deviation, use the symbol ±, not ( ) next to the mean value.

Response 11: Standard deviations are presented in parentheses as suggested by Strasak et al (2007): “In every case, standard deviations should preferably be reported in parentheses [ie, mean (SD)] than using mean ± SD expressions, as the latter specification can be confused with a 95% confidence interval by the reader”.

 Reference: Strasak AM, Zaman Q, Pfeiffer KP, Göbel G, Ulmer H. Statistical errors in medical research--a review of common pitfalls. Swiss Med Wkly. 2007 Jan 27;137(3-4):44-9.

Point 12:  In the beginning of “Results”:  I suppose the difference in gender prevalence is not significant, but yet, please, show the p-value.

Response 12: the significance of difference is tested when there are at least two variables. In this case, gender (a single variable) prevalence in total study population is presented.

Point 13:  Was there any gender difference in duration of treatment with warfarin?

Response 13: the mean treatment duration did not differ between males and females; a corresponding sentence has been added into “Results” section.

Point 14:  There is some repetition of data in the text before table 1 and in the very table 1 (for example for “gender”). The same for table 2. Please, avoid repetition of data in the text and the tables. 

Response 14: some of repetitive in-text information has been deleted or the corresponding sentences have been re-worded. However, we have left some of the most important of information in order to draw reader’s attention.

Point 15: Add also some p –values in tables 1 and 2, where relevant.

Response 15: the significance of difference is tested when there are at least two variables. In Tables 1 and 2, characteristics of total study population are presented.

Round 2

Reviewer 3 Report

The authors addressed the comments previously reported.